# Effect of Acupuncture on Oxidative Stress Induced by Cerebral Ischemia-Reperfusion Injury

**DOI:** 10.3390/antiox9030248

**Published:** 2020-03-19

**Authors:** Chao-Hsien Chen, Ching-Liang Hsieh

**Affiliations:** 1Department of Chinese Medicine, China Medical University Hospital, Taichung 40447, Taiwan; u100023024@cmu.edu.tw; 2Chinese Medicine Research Center, China Medical University, Taichung 40402, Taiwan; 3Graduate Institute of Acupuncture Science, College of Chinese Medicine, China Medical University, Taichung 40402, Taiwan

**Keywords:** acupuncture, oxygen free radical, ischemia-reperfusion injury

## Abstract

In this article, we review how acupuncture regulates oxidative stress to prevent ischemia–reperfusion injury. We electronically searched databases, including PubMed, Clinical Key and the Cochrane Library, from their inception to November 2019 by using the following medical subject headings and keywords: acupuncture, ischemia-reperfusion injury, oxidative stress, reactive oxygen species, and antioxidants. We concluded that acupuncture is effective in treating oxidation after ischemia-reperfusion injury. In addition to increasing the activity of antioxidant enzymes and downregulating the generation of reactive oxygen species (ROS), acupuncture also repairs the DNA, lipids, and proteins attacked by ROS and mediates downstream of the ROS pathway to apoptosis.

## 1. Introduction

Under normal conditions, a balance exists between free radical generation and elimination. If the balance is disturbed by the accumulation of free radicals or the dysfunction of free radical scavengers, an abundance of free radicals leads to cell injury or cell death, which is referred to as oxidative stress [1]. Free radicals are produced in both conditions of ischemia and ischemia-reperfusion [2]. Studies have revealed that a major cause of ischemia-reperfusion injury is reactive oxygen species (ROS) [3,4,5]. ROS, a type of free radical, are generated through aerobic metabolism. ROS are unstable compounds and easily react with deoxyribonucleic acid (DNA), lipids, and proteins and induce cell damage or cell death [6].

The brain is vulnerable to oxidative stress for several reasons, e.g., it consumes approximately 20% of the total basal oxygen and chemically transforms diverse reactive species to perform a variety of signaling functions [7].

Acupuncture, a branch of traditional Chinese medicine, is a form of therapy applied for more 3000 years in Asia. Acupuncture is practicing by medical doctor using the thin and sterile metal needles to insert into the human body at acupoint, the specific point based on the meridian theory of Chinese medicine, to perform stimulation [8]. Many previous studies have reported that acupuncture could treating plenty of disease such as ischemic stroke, musculoskeletal pain, postpartum depression, chronic urticarial, and cancer-related pain [9,10,11,12,13]. Furthermore, the effects and mechanisms of acupuncture for cerebral ischemia have been reported detailed nowadays. The mechanism of acupuncture therapy for cerebral ischemia includes improving the brain flows, inhibiting oxidative stress and inflammation, restoring the blood–brain barrier, promoting angiogenesis [14]. Electroacupuncture (EA), a combination of modern electric stimulation and acupuncture, have been reported to be effective for treating numerous nervous system disorders by affecting synaptic plasticity, elevating neurotrophic factors, offering neuroprotection, fostering maintenance of the blood–brain barrier, and inducing cell proliferation, antioxidant activity, anti-inflammatory and anti-apoptosis effects [15,16]. Low-intensity laser therapy has been used in neurorehabilitation and in cases of cognitive dysfunction and traumatic brain injury [17,18]. Moreover, laser acupuncture (LA), the combination of low-intensity laser therapy and acupuncture, has also been widely used for several disorders [19].

In this article, we review how acupuncture regulates oxidative stress to prevent cerebral ischemia-reperfusion injury.

## 2. Materials and Methods

We electronically searched several databases, including PubMed, Clinical Key, and the Cochrane Library from their inception to November 2019. We used the following Medical Subject Headings and keywords alone or in varied combinations: acupuncture, ischemia-reperfusion injury, oxidative stress, ROS, and antioxidants. In addition, we used Boolean operators (“not,” “and,” and “or”) to narrow or broaden the search results.

All articles written in English were manually screened, and relevant studies were identified. In the first step of selection criteria, we included 74 articles in Pubmed, 1 article in Cochrane and 58 articles in ClinicalKey, excluded 971 articles without abstract, not written in English, and not related to reperfusion injury or acupuncture, or ROS in abstract. In the second step of selection criteria, we included 44 articles and excluded 89 articles which are identical from different database, not available in full text, not related to reperfusion injury or acupuncture, or ROS in full text. The flowchart of our research process is presented in Figure 1.

## 3. Results

### 3.1. The Mechanism of ROS and Ischemia-Reperfusion Injury

Ischemia-reperfusion injury occurs in not only the brain but also the heart, skeletal muscles, skin, lungs, eyes, spinal cord, intestines, liver, kidneys, uterus, ovaries, testicles, penis, and joints. The mechanism and pathway of ROS and ischemia-reperfusion injury are not well understood, but evidence suggests that (1) enhancing ROS scavengers can reduce ischemia-reperfusion injury, (2) ROS generation and the cellular “footprints” can be detected in post-ischemia tissue, and (3) intervention involving ROS generation in normal tissue causes ischemia-reperfusion injury [5]. Different sources of ROS exist in different tissues. The main sources of ROS in the brain are NADPH oxidase (NOX), mitochondria, Xanthine oxidase (XO) and Monoamine oxidase (MAO) [5,20].

Ischemia stroke, cardia arrest, sepsis, and birth asphyxia are some of several causes of ischemia and hypoxia in the brain [21]. The occlusion of arterial blood flow causes hypoxia and leads to dysfunction of the electron transport chain in mitochondria, which results in insufficient ATP and the activation of anaerobic metabolism. Anaerobic metabolism causes lactate accumulation and contributes to metabolic acidosis [22]. Without sufficient ATP, the Na^+^-K^+^ and Ca^2+^ pumps are dysfunctional, which causes Na^+^ and Ca^2+^ retention in cells [21]. Na^+^ retention also reduces the activity of the Na^+^-H^+^ pump, causing a high level of H^+^ in cells and promoting acidosis. Metabolic acidosis may lead to impaired enzyme activity and the clumping of nuclear chromatin. The accumulation of H^+^, Na^+^, and Ca^2+^ causes an imbalance in osmolality, causing water to enter the cytoplasm [3].

In the reperfusion stage, oxygen availability after ischemia resets the mitochondrial respiratory chain, resulting in a large increase in superoxide radicals (O_2_^−^), mostly from complex I. Moreover, a high level of Ca^2+^ transforms into mitochondria and causes the mitochondrial permeability transition pore (MPTP) to open. MPTP governs the entry of water into the cytoplasm. The entry of water into the mitochondria causes swelling and damage, which may halt ATP synthase and generate more ROS [2,23]. Another source of ROS in the reperfusion stage is NOX, which is located in cell membranes. NOX enzymes use oxygen as the final electron acceptors through NADPH and immediately produce O_2_^−^. O_2_^−^ passes through the membrane via the anion channel pores, leading to nitric oxide (NO) degradation, peroxynitrite formation, and protein tyrosine nitration [3]. Xanthine oxidoreductase plays an important role of purine catabolism, which is a complex molybdoflavoenzyme containing xanthine dehydrogenase (XDH) and xanthine oxidase (XO). XDH uses NAD+ as an electron acceptor to oxidize hypoxanthine to xanthine, while XO uses O_2_ as the terminal electron acceptor to oxidize xanthine to uric acid [3,5]. Monoamine oxidases locate at the outer membrane of mitochondria and contribute to increase in H_2_O_2_ production from O_2_ and H_2_O and catecholamine release during brain ischemia and reperfusion [5,24]. There are two isoforms of MAO, MAO-A and MAO-B. MAO-A. The MAO-A mainly metabolizes the norepinephrine and 5-HT, whether the MAO-B enzyme mainly metabolizes phenylethylamine. Moreover, both of them break down dopamine equally [25]. MAO is mainly related to Parkinson disease and depression [26].

Above of all, there are several sources of ROS generation in the brain. ROS cause cell injury in different ways, lipid peroxidation, protein denaturation, DNA modifications and pathway to cell apoptosis and necrosis [2,22]. The product of protein denaturation depends on the different amino acid attacked by ROS. The side chain of amino acid residues attacked by ROS may lead to the dysfunction of the protein and the enzyme, and contributing to the alteration of membrane permeability [27]. ROS also cause DNA modifications by attacking both the purine and pyrimidine bases, and also the deoxyribose backbone, which participate in mutagenesis, carcinogenesis, and ageing [6].

### 3.2. Acupuncture Regulates Oxidative Stress

The following sections address means of combating oxidative stress, namely: (1) eliminating ROS with antioxidant enzymes or other signal pathway, (2) regulating the generation of ROS, (3) repairing proteins, lipids, or DNA that have been attacked by ROS, and (4) inhibiting downstream of the ROS pathway to cell apoptosis or autophagy [1].

#### 3.2.1. Eliminating ROS with Antioxidant Enzymes or Other Signaling Pathways

ROS are scavenged by free radical scavengers, also known as antioxidant enzymes. The following are common antioxidant enzymes in brain tissue [28]. Superoxide dismutase (SOD), the most crucial antioxidant enzyme, can be separated into three isoforms: copper zinc SOD (CuZnSOD), manganese SOD (MnSOD), and extracellular SOD (ECSOD). The three types of SOD are distributed in different locations. CuZnSOD is present in the cytoplasm and in the mitochondrial intermembrane space. MnSOD is present in the mitochondrial matrix. ECSOD is present in the extracellular space, cerebrospinal fluid, and cerebral vessels. All three isoforms can convert O_2_^−^ to H_2_O_2_ [23], and H_2_O_2_ can be converted to H_2_O and O_2_ by glutathione peroxidase (GPx). Catalase (CAT) has the same function as GPx, but because of its low activity in the adult brain and poor expression in neurons, it appears to play a secondary role to GPx in the scavenging of H_2_O_2_ [20].

ROS can cause cellular injury by lipid peroxidation. Malondialdehyde (MDA) and 4-hydroxynonenal (4-HNE), the end products of lipid peroxidation, are used to measure the extent of lipid peroxidation. MDA demonstrates specific reactivity on proteins, whereas 4-HNE has specificity on DNA and membranes, and both MDA and 4-HNE induce cell apoptosis [29].

A study revealed that EA at the Fengchi acupoint (GB20) can increase total SOD activity and GPx activity in rat brains with ischemic–reperfusion. Furthermore, EA reduced both MDA and MDA/4-HNE levels in rat brains with ischemic–reperfusion, which suggests that EA can reduce the extent of lipid peroxidation in brains with ischemic–reperfusion injury [30]. Another study reported that EA at GB20 or the Zusanli acupoint (ST36) before ischemia intervention can reduce MDA and MDA+4-HNE levels, respectively. No significant difference was observed between performing EA on GB20 and ST36 [29]. EA at the Baihui acupoint (GV20) and Dazhui acupoint (GV14) reportedly reduced MDA levels and increased SOD and GPx activity. In addition, it reduced not only the content of NO but also the activity of NO synthase. As mentioned, NO reacts with O_2_^−^ and generates ONOO-, a type of free radical that can inhibit cellular respiration by inhibiting the activity of the mitochondrial enzymes in addition to inhibiting DNA synthesis. All of these effects lead to cellular injury [31]. Another study reported that Acupuncture at GV20, Dazhui (GV14), Renzhong (GV26) and Fengfu (GV16) would also reduce the content of NO and the activity of NO synthase, and enhance the activity of SOD [32]. EA at the Tanzhong (CV17), Zhongwan (CV12), Qihai (CV6), ST36, and Xuehai (SP10) can increase the activity of SOD and GSH-Px and reduce expression of CuZnSOD mRNA in the hippocampus in cerebral multi-infarction rats. However, no difference was observed in CAT activity after EA [33]. Another study reported the similar result with the same acupoint. EA at the Tanzhong (CV17), Zhongwan (CV12), Qihai (CV6), ST36, and Xuehai (SP10) elevated the activities of total SOD, CuZnSOD, and MnSOD, decreased the level of MDA and superoxide anion, and regulated the ratio of reduced glutathione (GSH) and oxidized glutathione (GSSG) in mitochondria [28]. A different study used three types of electronic waves (continuous, dilatational, and intermittent) to determine which type of EA attenuates oxidative stress more effectively. All three types of EA on GV20 and ST36 reduced MDA levels and increased CAT and SOD activity in serum and the hippocampus. No significant difference was observed between the three types of EA [34]. Another study revealed that EA pretreatment upregulated neuronal expression of MnSOD. To determine the pathway precisely, MnSOD small interfering RNA and other agonists and antagonists of cannabinoid receptor type 1 receptor (CB1R) were added to the trail to confirm the pathway. The transcription factor directly responsible for MnSOD gene expression is the signal transducer and activator of transcription 3 (STAT3), and CB1R mediates STAT3 phosphorylation [35].

LA, a combination of acupuncture and low-intensity laser therapy, has been used to treat various conditions. In the present study, LA at GV20 reduced the MDA level and increased the activity of CAT and GPx in the cerebral cortex and SOD in the mitochondria in rats with occlusion of the right middle cerebral artery [36]. Another study reported similar results: LA elevated SOD and GPx activity in the hippocampus. In addition, LA also reduced the level of interleukin-6 (IL-6), the inflammation factor associated with inflammation-induced neurotoxicity [37].

Nrf2 is a transcription factor that is sensitive to redox. Activated Nrf2 reacts with antioxidant response element to mediate anti-oxidative action. EA at GV20 and GV14 could increase the expression of Nrf2 in rats with middle cerebral artery occlusion [38]. Another study reported that acupuncture at GV20 and ST36 enhance nuclear translocation of Nrf2 in neurons and up-regulate the protein and mRNA levels of Nrf2 [39]. The abovementioned results are summarized in Table 1.

#### 3.2.2. Regulating the Generation of ROS

NOX is a major ROS-producing enzyme that produces O_2−_ in the brain under certain physiological conditions. EA stimulation at GV20 and ST36 could suppress NOX-derived O_2−_ generation and reduce the expression of NOX subunits [40].

Pretreatment EA at GV20 and GV14 was reported to downregulate the expression of NOX4, one isoform of NOX [41]. Furthermore, pretreatment EA at GV20 in diabetic mice with cerebral ischemia not only suppressed activation of NOX after reperfusion but also reduced MDA content and ROS formation after reperfusion [42]. The aforementioned results are summarized in Table 2.

#### 3.2.3. Repairing Proteins, Lipids, or DNA That Have Been Attacked by ROS

ROS may attack mitochondrial respiratory enzymes, including cytochrome c oxidase, succinic dehydrogenase, and NADH dehydrogenase. EA at the Renzhong acupoint (GV26) and GV20 on focal cerebral ischemia in rats enhanced mitochondrial respiratory enzymatic activity. EA also improved mitochondrial respiratory function directly or indirectly by increasing the activity of respiratory enzymes [43]. As mentioned, ROS may induce oxidative DNA damage. Redox effector factor (Ref-1) is a multifunctional protein involved in base excision DNA repair of apurinic/apyrimidinic sites in DNA, which is a common form of DNA damage after oxidative stress [44]. EA stimulation increases the expression of Ref-1 in the hippocampus, which suggests that it can reduce oxidative-induced damage to DNA [45]. The aforementioned results are summarized in Table 3.

#### 3.2.4. Inhibiting Downstream of the ROS Pathway to Cell Apoptosis or Autophagy

P38/MAPK mediate various cellular signaling pathways in response to extracellular apoptotic stimuli and intracellular oxidative stress after transient focal cerebral ischemia [46]. P38/MAPK activation upregulates the expression of the cAMP response element-binding protein (CREB), which regulates the B-cell lymphoma 2 (Bcl-2) family [47]. The Bcl-2 family separates into two groups, the pro-apoptosis group (Bax: Bcl-2 associated X and Bad: Bcl-2-associated death promoter) and anti-apoptosis group (Bcl-2 and Bcl2-xl: B-cell lymphoma extralarge) [48]. Activation of CREB phosphorylation can increase Bcl-2 expression, leading to the protection of the cell. In the present study, EA at GV20 and Shenting (GV24) increases the reactivity of p-CREB and the expression of Bcl-2, and decreases the expression of Bax. There are also increasing activity of the antioxidant enzymes SOD and GPx and decreasing MDA content [47]. Another study reported the similar result as the previous study that EA at GV20 and GV24 could increase the level of Bcl-2 and reduce the expression of Bax [49]. Another study reported that EA at the Chize acupoint (LU5), Hegu acupoint (LI4), ST36, and SP6 could inhibit the expression of phosphorylated p38 MAPK in the CA1 area of the hippocampus [46]. Another study also reported that EA at GV20 and GV24 increased the Bcl-2/Bax ratio [50]. In addition, EA pretreatment at not only prevents P38/MAPK activation, but also increased the activities of GSH and SOD and downregulated the levels of MDA [51].

The pro-apoptosis group in Bcl-2 family led to the reduction in activations of caspase 3 and caspase 9, causing caspases cascade and apoptosis. In the present study, acupuncture at GV20 and Si shencong (Ex-HN1) down-regulates the expression of Bax and the activation of caapase 3 and caspase 9, and up-regulated levels of Bcl-2 [52]. Another study reported that EA at GV20 and the Fengfu acupoint (GV16) suppressed the p38 MAPK-mediated antiapoptotic signaling pathways. Furthermore, the study also reported that EA also increases the ratios of mitochondrial Bcl-xL/Bax and Bcl-2/Bax ratio, and suppressed caspase-3 activity [48].

Another pathway, Phosphoinositide 3-kinase/protein kinase B (PI3K/Akt) signaling pathway, also regulates the cell survival metabolism. Activation of PI3K/Akt pathway would inhibit cell apoptosis. The present study reported that EA at ST36 and LI11 increases the level of Bcl-2 and stimulates the PI3K/Akt pathway, and it also suppresses the expression of Bax and cleaved Caspase-3 [53]. Another study also reported the same result that EA at ST36 and LI11 elevated the expression of PI3K/Akt pathway and Bcl-2, and it also inhibit the expression of Bax [54].

Beclin-1, which is indispensable to the recruitment of other autophagic proteins during the expansion of pre-autophagosomal membrane, and microtubule-associated protein light chain 3 (LC3) proteins increased in cerebral cortex, which indicated that autophagy was activated after the cerebral ischemia-reperfusion injury [55]. There are two type of form of LC3, cytosolic form (LC3-I, LC3A) and membrane-bound form (LC3-II, LC3B). The LC3-I binds to phosphatidylethanolamine to form LC3-II, which is the specific marker used as autophagy-induced subject matter. [3,55] In the present study, EA at GV20 pretreatment decreases the expression of LC3-II and the ratio of LC3-II/LC3-I, also suppresses the expression of Beclin-1 [55]. Another study reported similar results. EA at GV26 reduces the expression of Beclin-1 and LC3-II, and reduced the number of apoptosis cell, and elevated the level of Bcl-2 [56].

With the lack of nutrients and energy, cells display inhibited activity of mTOR, while ULK1 is activated, and then control Atg13 and RB1CC, autophagy membrane is ready to generate [3,57]. The present study reported that EA at the LI11 and ST36 could decrease the level of LC3BII/LC3BI, ULK1, Atg13 and Beclin1, and increase the expression of mTOR complex 1 (mTORC1), which is an active form of mTOR [58]. Another study reported that EA at GV20, ST36, and Mingmen (GV4) decreased levels of LC3 and Beclin1 and elevated levels of mTOR. Furthermore, EA also reduced the contents of MDA and elevated the vitality of SOD [57].

The neuroprotection against focal cerebral ischemia provided by various agents is through the upregulation of Hypoxia inhibitory factor-α (HIF-1α). HIF-1α regulates the expression of its gene, heme oxygenase-1 (HO-1). In the present study, EA pretreatment at GV20 upregulated the expression of Bcl-2, HIF-1α, and HO-1 and down-regulated the expression of Bax [59]. The aforementioned results are summarized in Table 4.

## 4. Discussion

ROS is a fundamental mechanism of brain damage in stroke and reperfusion ensuing stroke. There are different types of damage by ROS, such as lipid peroxidation, protein denaturation, DNA modifications. Regardless of the type of damage, there are variable ways for acupuncture to counteract the damage by ROS. Acupuncture not only regulates the headstream of ROS pathway to inhibit the formation of ROS, but also regulates the downstream to repair the damage by ROS and inhibits the ROS pathway to cell apoptosis and autophagy.

Several studies have reported that CAT activity significantly increased as a result of EA treatment, whereas other studies observed no significant difference in CAT activity after EA treatment and attributed this finding to the low brain activity in rats [33,34,36]. Another study revealed that CAT exhibits low activity in the human brain and poor expression in neurons [20]. However, the controversial outcomes of these trials have yet to be satisfactorily explained. Several trials on rats involved an acupuncture intervention before arterial occlusion, whereas others involved acupuncture intervention after arterial occlusion [29,30,33,36]. All experiments have revealed that acupuncture or EA is beneficial to reducing oxidative stress, but neither approach has been highlighted as more effective than the other. These results should eventually be applied in clinical treatment. We believe that the results of acupuncture intervention after arterial occlusion on rats suggest that this method is appropriate for clinical use. For future studies, researchers could consider comparing the effects of traditional acupuncture, EA, or LA on combating oxidative stress. Studies have revealed no significant difference between different acupoints, EA frequency, or type of EA wave. Numerous variables are present in the three types of acupuncture such as the retention time of the acupuncture, the frequency of treatment, and the power and wavelength of LA.

All of the trials we reviewed in this article were conducted with animal models, but there may be some differences between rat and human brains that would influence the results, such as different antioxidant enzyme concentrations or reactions to acupuncture.

## 5. Conclusions

Acupuncture effectively treats oxidation after ischemia-reperfusion injury. In addition to increasing the activity of antioxidant enzymes and downregulating the generation of ROS, acupuncture also repairs the DNA, lipids, and proteins attacked by ROS and mediates downstream of the ROS pathway to apoptosis or autophagy.

## Figures and Tables

**Figure 1 antioxidants-09-00248-f001:**
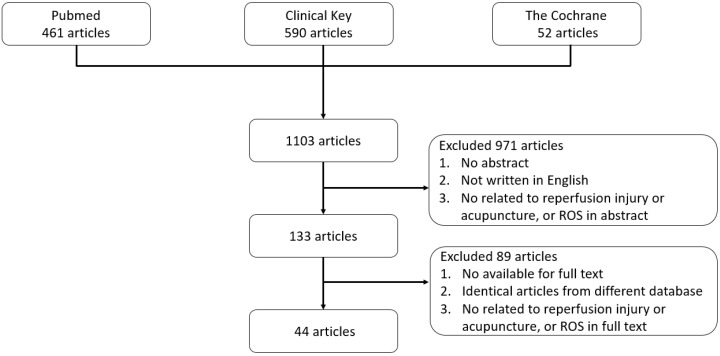
Flowchart of the search process.

**Table 1 antioxidants-09-00248-t001:** Acupuncture eliminates ROS with antioxidant enzymes or other signaling pathways.

References	Species/Model	Acupuncture Type/Frequency/Intensity/Time	Acupoints	Results
Siu et al., 2004 [30]	SD rats/MCAo	EA/2 Hz/0.7 V/30 min	Fengchi (GB20)	Increased total SOD and GPx activity Reduced both MDA and MDA/4-HNE levels
Siu et al., 2004 [29]	SD rats/MCAo	EA/2 Hz/0.7 mV/30 min	Fengchi (GB20) or Zusanli (ST36)	Reduced the amount of MDA and MDA+4-HNE No significant difference between EA on GB20 and ST36
Wang et al., 2004 [31]	SD rats/4-vessel occlusion method	EA/150 Hz/2 mA/20 min	Baihui (GV20) Dazhui (GV14)	Increased SOD and GPx activity Reduced the level of MDA Reduced the content of nitric oxide (NO) Reduced the activity of NO synthase
Su et al., 2019 [32]	SD rats/MCAo	TA/twisted/20 min	Baihui (GV20) Dazhui (GV14) Renzhong (GV26) Fengfu (GV16)	Increased SOD activity Reduce the content of NO and the activity of NO synthase
Liu et al., 2006 [33]	Wistar rats/clip at ECA	TA/twisted twice per second at each point/30 s	Tanzhong (CV17) Zhongwan (CV12) Qihai (CV6) Zusanli (ST36 Xuehai (SP10)	Increased the activity of SOD and GPx Reduced the expression of CuZnSOD mRNA in the hippocampus No significant difference in CAT activity
Zhang et al., 2014 [28]	Wistar rats/clip at ECA	TA/twisted twice per second at each point/30 s	Tanzhong (CV17) Zhongwan (CV12) Qihai (CV6) Zusanli (ST36) Xuehai (SP10)	Increased the activities of total SOD, CuZnSOD and MnSOD, Reduced the level of MDA, GSH and GSSG in mitochondria.
Chen et al., 2016 [34]	SD rats/cecum with ligation (simulating sepsis)	CW: EA/2 Hz/2 mA/30 min IW: EA/2/0 Hz/2 mA/30 min DW: EA/2/15 Hz/2 mA	Baihui (GV20) Zusanli (ST36)	Increased CAT and SOD activity in serum and the hippocampus Reduced MDA levels in serum and the hippocampus No significant difference between three types of EA wave
Sun et al., 2016 [35]	C57BL/6 mice/MCAo	EA/2/15 Hz/1 mA/30 min	Baihui (GV20)	Upregulated the neuronal expression of MnSOD
Jittiwat, J., 2017 [36]	Wistar rats/MCAo	LA/810 nm laser beam/100 mW, as pulsed wave (50%)/10 min	Baihui (GV20)	Reduced MDA level Increased CAT and GPx activity in the cerebral cortex Increased SOD activity in mitochondria
Jittiwat, J., 2017 [37]	Wistar rats/MCAo	LA/810 nm laser beam/100 mW, as pulsed wave (50%)/10 min	Baihui (GV20)	Elevated SOD and GPx activity in the hippocampus Reduced IL-6 level
Shen et al., 2016 [38]	SD rats/MCAo	EA/2/15 Hz/1–3 mA/30 min	Baihui (GV20) Dazhui (GV14)	Increased the expression of Nrf2
Wang et al., 2015 [39]	Wister rats/bilateral CCAo	TA/unkwoun	Baihui (GV20) Zusanli (ST36)	Increased nuclear translocation of Nrf2 in neurons Increased the protein and mRNA levels of Nrf2

Abbreviations: ROS: reactive oxygen species; SD rats: Sprague–Dawley rats; MCAo: middle cerebral artery occlusion; ECA: external carotid artery; EA: electroacupuncture; TA: traditional acupuncture; LA: laser acupuncture; CW: continuous wave; IW: intermittent wave; DW: dilatational wave; GSSG: oxidized glutathione; CCAo: common carotid artery occlusion.

**Table 2 antioxidants-09-00248-t002:** Acupuncture regulates the generation of ROS.

References	Species/Model	Acupuncture Type/Frequency/Intensity/Time	Acupoints	Results
Shi et al., 2015 [40]	Wistar rats/bilateral CCAo	TA/twirling reinforcing manipulation >2 Hz for each point/30 s	Baihui (GV20) Zusanli (ST36)	Suppressed NADPH oxidase-derived O_2_^−^ generation Reduced the expression of NADPH oxidase subunits
Jung et al., 2016 [41]	C57/BL6J mice/MCAo	EA/2 Hz/2V/20min	Baihui (GV20) Dazhui(GV14)	Downregulated the expression of NOX4
Gou et al., 2014 [42]	C57/BL6J mice/DM + MCAo	EA/2/15 Hz/1 mA/30 min	Baihui (GV20)	Suppressed activation of NADPH oxidase Reduced MDA content and ROS formation

Abbreviation note: ROS: reactive oxygen species; TA: traditional acupuncture; CCAo: common carotid artery occlusion; DM: diabetes mellitus; MCAo: middle cerebral artery occlusion.

**Table 3 antioxidants-09-00248-t003:** Acupuncture repairs proteins, lipids, or DNA that have been attacked by ROS.

References	Species/Model	Acupuncture Type/Frequency/Intensity/Time	Acupoints	Results
Zhong et al., 2007 [43]	SD rats/MCAo	EA/5/20 Hz/2–4 mA/60 min	Baihui (GV20) Renzhong (GV26)	Enhanced mitochondrial respiratory enzymatic activity Improved mitochondrial respiratory function
Liu et al., 2013 [45]	Wistar rats/ICAo	TA/twisted at the speed of twice per second in each point/30 s	Tanzhong (CV17) Zhongwan (CV12) Qihai (CV6) Zusanli (ST36) Xuehai (SP10)	Increased the expression of Ref-1 in the hippocampus

Abbreviation note: ROS: reactive oxygen species; SD rats: Sprague–Dawley rats; MCAo: middle cerebral artery occlusion; ICAo: internal carotid artery occlusion; TA: traditional acupuncture.

**Table 4 antioxidants-09-00248-t004:** Acupuncture inhibits downstream of the ROS pathway to cell apoptosis or autophagy.

References	Species/Model	Acupuncture Type/Frequency/Intensity/Time	Acupoints	Results
Lin et al., 2015 [47]	SD rats/MCAo	EA/5/20 Hz/1–3 mA/30 min	Baihui (GV20) Shenting (GV24)	Increased the reactivity of p-CREB and expression of Bcl-2 Reduced the expression of Bax Increased the activity of the SOD and GPx Reduced MDA content
Liu et al., 2018 [49]	SD rats/MCAo	EA/1/20 Hz/1 mA/30 min	Baihui (GV20) Shenting (GV24)	Increased the level of Bcl-2 Reduced the expression of Bax
Lan et al., 2017 [46]	SD rats/MCAo	EA/2/50 Hz/unknown/20 min	Chize (LU5 Hegu (LI4) Zusanli (ST36) Sanyinjiao (SP6)	Inhibited the expression of phosphorylated p38 MAPK in the CA1 area of the hippocampus
Liu et al., 2015 [50]	SD rats/MCAo	EA/1/20Hz/6mA/30 min	Baihui (GV20) Shenting (GV24)	Increased the Bcl-2/Bax ratio
Long et al., 2019 [51]	SD rats/MCAo	EA/2/100 Hz/1 mA/10 min with EAand 5 min without electricalstimulation	Baihui (GV20) Shenshu (BL23) Sanyinjiao (SP6)	Prevented P38/MAPK activation Increased the activities of GSH and SOD Reduced the levels of MDA.
Zhang et al., 2015 [52]	SD rats/MCAo	TA/twisted at the speed of twice per second for 15 s/30 min	Baihui (GV20) Sishencong (Ex-HN1)	Increased levels of Bcl-2 Reduced the expression of Bax Reduced the activation of caapase 3 and caspase 9
Cheng et al., 2015 [48]	SD rats/MCAo	EA/5 Hz/2.7–3 mA/25 min EA/25 Hz/2.7–3 mA/25 min	Baihui (GV20) Fengfu (GV16)	Increased the ratios of mitochondrial Bcl-xL/Bax and Bcl-2/Bax ratio Suppressed the p38 MAPK-mediated antiapoptotic signaling pathways and caspase-3 activity
Xue et al., 2014 [53]	SD rats/MCAo	EA/4/20 Hz/unknown/30 min	Zusanli (ST36) Quchi (LI11)	Increased the level of Bcl-2 Stimulated the PI3K/Akt pathway Suppressed the expression of Bax and cleaved Caspase-3
Chen et al., 2012 [54]	SD rats/MCAo	EA/1/20 Hz/unknown/30 min	Zusanli (ST36) Quchi (LI11)	Elevated the expression of PI3K/Akt pathway and Bcl-2 Inhibited the expression of Bax
Wu et al., 2014 [55]	SD rats/MCAo	EA/2/15 Hz/1 mA/30 min	Baihui (GV20)	Reduced the expression of LC3-II and Beclin-1, and the ratio of LC3-II/LC3-I
Shu et al., 2016 [56]	SD rats/MCAo	EA/2/20 Hz/unknown/30 min	Renzhong (GV26)	Increased the level of Bcl-2 Reduced the expression of Beclin-1 and LC3-II Reduced the number of apoptosis cell
Ting et al. 2017 [57]	SD rats/4-vessel occlusion method	EA/40/50Hz/unknown/unknown	Baihui (GV20) Zusanli (ST36) Mingmen (GV4)	Elevated level of mTOR and the vitality of SOD Decreased levels of LC3 and Beclin1, and MDA
Liu et al., 2016 [58]	SD rats/MCAo	EA/1/20 Hz/0.2 mA/30 min	Quchi (LI11) Zusanli (ST36)	Increased the expression of mTOR complex 1 Reduced the level of LC3BII/LC3BI, ULK1, Atg13 and Beclin1
Zhao et al., 2015 [59]	SD rats/MCAo	EA/2/15 Hz/1 mA/30 min	Baihui (GV20)	Increased the expression of Bcl-2, HIF-1α and HO-1 Reduced the expression of Bax

Abbreviations: ROS: reactive oxygen species; SD rats: Sprague–Dawley rats; MCAo: middle cerebral artery occlusion.

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
