# Peer review of "Effect of Acupuncture on Oxidative Stress Induced by Cerebral Ischemia-Reperfusion Injury"

_antioxidants, 2020, doi:10.3390/antiox9030248_

Round 1

Reviewer 1 Report

In this paper the Authors, through a scrupulous literature review work demonstrate how acupucture increases the activity of antioxidant enzymes and downregulates the generation of reactive oxygen species (ROS); also acupuncture seems to repair the ROS-dependent DNA, lipids, and protein damage.

The data lend support to the hypothesis that acupuncture might represent a new tool to counteract the cerebral-ischemia reperfusion-dependent alterations.

Considering that the main cause of these alterations is the formation of ROS, one could assume that, irrespective of the type of damage, all the alterations that are caused by the increase in ROS can be counteracted by the use of acupuncture. I suggest that this aspect could be commented on in the discussion.

Author Response

#Reviewer 1

  1. In this paper the Authors, through a scrupulous literature review work demonstrate how acupuncture increases the activity of antioxidant enzymes and downregulates the generation of reactive oxygen species (ROS); also acupuncture seems to repair the ROS-dependent DNA, lipids, and protein damage.

#RESPONSE:

  • Thank you very much for the reviewer’s comments.
  1. The data lend support to the hypothesis that acupuncture might represent a new tool to counteract the cerebral-ischemia reperfusion-dependent alterations.

#RESPONSE:

  • Thank you very much for the reviewer’s comments.
  1. Considering that the main cause of these alterations is the formation of ROS, one could assume that, irrespective of the type of damage, all the alterations that are caused by the increase in ROS can be counteracted by the use of acupuncture. I suggest that this aspect could be commented on in the discussion.

#RESPONSE:

  • Thank you very much for the reviewer’s comments.
  • Discussion:

“ROS is a fundamental mechanism of brain damage in stroke and reperfusion ensuing stroke. There are different types of damage by ROS, such as lipid peroxidation, protein denaturation, DNA modifications. Regardless of the type of damage, there are variable ways for acupuncture to counteract the damage by ROS. Acupuncture not only regulates the headstream of ROS pathway to inhibit the formation of ROS, but also regulates the downstream to repair the damage by ROS and inhibits the ROS pathway to cell apoptosis and autophagy.” had been added at Line 269-274.

Reviewer 2 Report

This paper aims to review the effect of acupuncture on reoxygenation-induced ROS damage. The study is based on 21 papers from different sources, part of which in Chinese, which are not easily accessible to a fraction of readers of Antioxidants. Out of 1154 initial papers, only 21 were used. The selection criteria are not described in sufficient detail; e.g. using the selection criteria yields 42 titles with abstracts in Pubmed alone. A basic knowledge of acupuncture is needed to understand the paper, and the description of ROS formation is rather simple, e.g. monoamine oxidases and protein radicals are not mentioned are not mentioned. The subject deserves a more comprehensive review of the subject.

Author Response

#Reviewer 2

  1. This paper aims to review the effect of acupuncture on reoxygenation-induced ROS damage. The study is based on 21 papers from different sources, part of which in Chinese, which are not easily accessible to a fraction of readers of Antioxidants.

#RESPONSE:

  • Thank you very much for the reviewer’s comments.
  • As the reviewer’s comments Chinese is not easily accessible to a fraction of readers of antioxidants.
  • The data of the manuscript is from PubMed research, we selected “references” only in “English”.
  1. Out of 1154 initial papers, only 21 were used. 

#RESPONSE:

  • Thank you very much for the reviewer’s comments.
  • As the reviewer’s comments, only 21 in 1154 papers were used , the number is small, therefore, we added 26 articles again from PubMed in introduction and 1, 3.2.1 and 3.2.4 of Results section as “highlight in red color”.
  1. The selection criteria are not described in sufficient detail; e.g. using the selection criteria yields 42 titles with abstracts in Pubmed alone.

#RESPONSE:

  • Thank you very much for the reviewer’s comments.

(2) As the reviewer’s comment, the selection criteria need description.

(3) Materials and Methods:

“We electronically searched several databases, including PubMed, Clinical Key, the Cochrane Library, and the China National Knowledge Infrastructure, from their inception to November 2019. We used the following Medical Subject Headings and keywords alone or in varied combinations: acupuncture, ischemia–reperfusion injury, oxidative stress, ROS, and antioxidants. In addition, we used Boolean operators (“not,” “and,” and “or”) to narrow or broaden the search results.

All articles written in English or Chinese were manually screened, and relevant. studies were identified. Articles for which the full text was unavailable, those written in other languages, those unrelated to neuroprotection and acupuncture according to the abstract, and those unrelated to neuroprotection and acupuncture in the full text were excluded. The flowchart of our research process is presented in Figure 1.” had been changed to “We electronically searched several databases, including PubMed, Clinical Key, and the Cochrane Library from their inception to November 2019. We used the following Medical Subject Headings and keywords alone or in varied combinations: acupuncture, ischemia–reperfusion injury, oxidative stress, ROS, and antioxidants. In addition, we used Boolean operators (“not,” “and,” and “or”) to narrow or broaden the search results.

All articles written in English were manually screened, and relevant studies were identified. In the first step of selection criteria, we included 74 articles in Pubmed, 1 article in Cochrane and 58 articles in ClinicalKey, excluded 971 articles without abstract, not written in English, and not related to reperfusion injury or acupuncture, or ROS in abstract. In the second step of selection criteria, we included 44 articles and excluded 89 articles which are identical from different database, not available in full text, not related to reperfusion injury or acupuncture, or ROS in full text. The flowchart of our research process is presented in Figure 1.” had been added at Line 55-66.

  1. A basic knowledge of acupuncture is needed to understand the paper,

#RESPONSE:

  • Thank you very much for the reviewer’s comments.
  • As the reviewer’s comments, A basic knowledge of acupuncture is needed to understand the paper.
  • Introduction:

“Acupuncture, a branch of traditional Chinese medicine, is a form of therapy applied for more 3000 years in Asia. Acupuncture is practicing by medical doctor using the thin and sterile metal needles to insert into the human body at acupoint, the specific point based on the meridian theory of Chinese medicine, to perform stimulation. Many previous studies have reported that acupuncture could treating plenty of disease such as ischemic stroke, musculoskeletal pain, postpartum depression, chronic urticarial and cancer-related pain. Furthermore, the effects and mechanisms of acupuncture for cerebral ischemia have been reported detailed nowadays. The mechanism of acupuncture therapy for cerebral ischemia includes improving the brain flows, inhibiting oxidative stress and inflammation, restoring the blood–brain barrier, promoting angiogenesis.” had been added at lines 35-44.

  1. and the description of ROS formation is rather simple, e.g. monoamine oxidases (MAO) and protein radicals are not mentioned.

#RESPONSE:

  • Thank you very much for the reviewer’s comments.
  • As the reviewer’s comments, the description of ROS formation is rather simple, e.g. monoamine oxidases (MAO) and protein radicals are not mentioned.
  • Results:

“The main sources of ROS in the brain are NADPH oxidase (NOX), mitochondria, Xanthine oxidase(XO) and Monoamine oxidase (MAO).” had been added at lines 75-76.

“Xanthine oxidoreductase plays an important role of purine catabolism, which is a complex molybdoflavoenzyme containing xanthine dehydrogenase(XDH) and xanthine oxidase(XO). XDH uses NAD+ as an electron acceptor to oxidize hypoxanthine to xanthine, while XO uses O2 as the terminal electron acceptor to oxidize xanthine to uric acid. Monoamine oxidases locate at the outer membrane of mitochondria and contribute to increase in H2O2 production from O2 and H2Oand catecholamine release during brain ischemia and reperfusion. There are two isoforms of MAO, MAO-A and MAO-B. MAO-A. The MAO-A mainly metabolizes the norepinephrine and 5-HT, whether the MAO-B enzyme mainly metabolizes phenylethylamine . And both of them break down dopamine equally. MAO is mainly related to Parkinson disease and depression. Above of all, there are several sources of ROS generation in the brain. ROS cause cell injury in different ways, lipid peroxidation, protein denaturation, DNA modifications and pathway to cell apoptosis and necrosis. The product of protein denaturation depends on the different amino acid attacked by ROS. The side chain of amino acid residues attacked by ROS may lead to the dysfunction of the protein and the enzyme, and contributing to the alteration of membrane permeability. ROS also cause DNA modifications by attacking both the purine and pyrimidine bases and also the deoxyribose backbone, which participate in mutagenesis, carcinogenesis, and ageing.” had been added at Line 94-111

  1. The subject deserves a more comprehensive review of the subject.

#RESPONSE:

  • Thank you very much for the reviewer’s comments.
  • As the reviewer’s comments, the subject deserves a more comprehensive review of the subject.
  • Results: 3.2.4

“P38/MAPK activation upregulates the expression of the cAMP response element-binding protein (CREB), which regulates the B‑cell lymphoma 2 (Bcl‑2) family.[47] The Bcl-2 family separates into 2 groups, the pro-apoptosis group (Bax: Bcl-2 associated X and Bad: Bcl-2-associated death promoter)and anti-apoptosis group (Bcl-2 and Bcl2-xl: B-cell lymphoma extralarge).[48] Activation of CREB phosphorylation can increase Bcl-2 expression, leading to the protection of the cell. In the present study, EA at GV20 and Shenting (GV24) increases the reactivity of p‑CREB and the expression of Bcl-2, and decreases the expression of Bax. There are also increasing activity of the antioxidant enzymes SOD and GPx and decreasing MDA content.[47] Another study reported the same result as the previous study that EA at GV20 and GV24 could increase the level of Bcl-2 and reduce the expression of Bax.[49]

 Another study also reported that EA at GV20 and GV24 increased the Bcl-2/Bax ratio.[50] In addition, EA pretreatment at not only prevents P38/MAPK activation, but also increased the activities of GSH and SOD and downregulated the levels of MDA.[51]

The pro-apoptosis group in Bcl-2 family led to the reduction in activations of caspase 3 and caspase 9, causing caspases cascade and apoptosis. In the present study, acupuncture at GV20 and Si shencong (Ex-HN1) down-regulates the expression of Bax and the activation of caapase 3 and caspase 9, and up-regulated levels of Bcl-2. [52] Another study reported that EA at GV20 and the Fengfu acupoint (GV16) suppressed the p38 MAPK-mediated antiapoptotic signaling pathways [48].

Another pathway, Phosphoinositide 3-kinase/protein kinase B (PI3K/Akt) signaling pathway, also regulates the cell survival metabolism. Activation of PI3K/Akt pathway would inhibit cell apoptosis. The present study reported that EA at ST36 and LI11 increases the level of Bcl-2 and stimulates the PI3K/Akt pathway, and it also suppresses the expression of Bax and cleaved Caspase-3.[53] Another study also reported the same result that EA at ST36 and LI11 elevated the expression of PI3K/Akt pathway and Bcl-2, and it also inhibit the expression of Bax.[54]

Beclin-1, which is essential for recruitment of other autophagic proteins during the expansion of pre‑autophagosomal membrane, and microtubule‑associated protein light chain 3 (LC3) proteins increased in cerebral cortex, which indicated that autophagy was activated after the cerebral ischemia-reperfusion injury.[55] There are two type of form of LC3, cytosolic form (LC3‑I, LC3A) and membrane‑bound form (LC3‑II, LC3B). The LC3-I binds to phosphatidylethanolamine to form LC3-II, which is the specific marker used as autophagy-induced subject matter. [3,55] In the present study, EA at GV20 pretreatment decreases the expression of LC3-II and the ratio of LC3-II/LC3-I, also suppresses the expression of Beclin-1.[55] Another study reported the similar results. EA at GV26 reduces the expression of Beclin-1 and LC3-II, and reduced the number of apoptosis cell, and elevated the level of Bcl-2.[56]

Mechanistic target of rapamycin (mTOR) is located downstream of PI3K/Akt signaling, regulates cell growth, and inhibits the initial process of cell autophagy. [57] In the absence of nutrients and energy, cells display inhibited mTOR activity, while ULK1 is activated, and then control Atg13 and RB1CC, autophagy membrane began to form. The present study reported that EA at the LI11 and ST36 could decrease the level of LC3BII/LC3BI, ULK1, Atg13 and Beclin1, and increase the expression of mTOR complex 1 (mTORC1), which is an active form of mTOR. [58]

The neuroprotection against focal cerebral ischemia provided by various agents is through the upregulation of Hypoxia inhibitory factor-α (HIF-1α). HIF-1α regulates the expression of its gene, heme oxygenase-1 (HO-1). In the present study, EA pretreatment at GV20 upregulated the expression of Bcl-2, HIF-1α and HO-1and down-regulated expression of Bax. [59]” had been added at Line 191-241.

#PS

  1. We added “26 references” as highlight in red color.
  2. “Figure 1” had been revised highlight in red color.
  3. 3. “Tables 1 and 4” had been revised as highlight in red color.

Round 2

Reviewer 2 Report

The revised ms improved considerable. The revised parts require spell check.